# Environmental morphing enables informed dispersal of the dandelion diaspore

Madeleine Seale[1,2,3,4], Oleksandr Zhdanov[5,6], Merel B Soons[7], Cathal Cummins[1,2,8†, ‡], Erika Kroll[1], Michael R Blatt[6], Hossein Zare-Behtash[5], Angela Busse[5], Enrico Mastropaolo[3§], James M Bullock[9], Ignazio M Viola[8], Naomi Nakayama[1,2,10,11]*

[1]School of Biological Sciences, Institute of Molecular Plant Sciences, University of Edinburgh, Edinburgh, United Kingdom; [2]Centre for Synthetic and Systems Biology, University of Edinburgh, Edinburgh, United Kingdom; [3]School of Engineering, Institute for Integrated Micro and Nano Systems, University of Edinburgh, Edinburgh, United Kingdom; [4]Department of Plant Sciences, University of Oxford, Oxford, United Kingdom; [5]James Watt School of Engineering, University of Glasgow, Glasgow, United Kingdom; [6]Laboratory of Plant Physiology and Biophysics, Bower Building, University of Glasgow, Glasgow, United Kingdom; [7]Ecology & Biodiversity group, Utrecht University, Utrecht, Netherlands; [8]School of Engineering, Institute for Energy Systems, University of Edinburgh, Edinburgh, United Kingdom; [9]UK Centre for Ecology & Hydrology, Wallingford, United Kingdom; [10]Centre for Science at Extreme Conditions, University of Edinburgh, Edinburgh, United Kingdom; [11]Department of Bioengineering, Imperial College London, South Kensington, United Kingdom

**\*For correspondence:**
n.nakayama@imperial.ac.uk

**Present address:** †Maxwell Institute for Mathematical Sciences, Department of Mathematics and Institute for Infrastructure and Environment, Heriot-Watt University, Edinburgh, United Kingdom; ‡School of Energy, Geosciences, Infrastructure and Environment, Heriot-Watt University, Edinburgh, United Kingdom

§Deceased

**Competing interest:** The authors declare that no competing interests exist.

**Abstract** Animal migration is highly sensitised to environmental cues, but plant dispersal is considered largely passive. The common dandelion, *Taraxacum officinale*, bears an intricate haired pappus facilitating flight. The pappus enables the formation of a separated vortex ring during flight; however, the pappus structure is not static but reversibly changes shape by closing in response to moisture. We hypothesised that this leads to changed dispersal properties in response to environmental conditions. Using wind tunnel experiments for flow visualisation, particle image velocimetry, and flight tests, we characterised the fluid mechanics effects of the pappus morphing. We also modelled dispersal to understand the impact of pappus morphing on diaspore distribution. Pappus morphing dramatically alters the fluid mechanics of diaspore flight. We found that when the pappus closes in moist conditions, the drag coefficient decreases and thus the falling velocity is greatly increased. Detachment of diaspores from the parent plant also substantially decreases. The change in detachment when the pappus closes increases dispersal distances by reducing diaspore release when wind speeds are low. We propose that moisture-dependent pappus-morphing is a form of informed dispersal allowing rapid responses to changing conditions.

## Editor's evaluation

This boundary-crossing work on dandelion diaspore flight is an excellent demonstration of how to address fundamental questions about wind dispersal of plant seeds from biophysical and ecological perspectives. Both wind-tunnel experiments and models provide compelling evidence that the aerodynamics of dandelion diaspores change with the environment. Addition of local climate data enables the authors to make a convincing case about how the biophysical properties can scale up

to affect dispersal across the landscape under different environmental conditions. In addition to the strong data, this is a clear, accessible, and very enjoyable read.

## Introduction

Long-distance dispersal to a new habitat is one of the riskiest decisions a living organism can make. From ballooning spiders dispersing from salt marshes according to local wind velocity (*Bonte et al., 2007*) to kittiwakes abandoning breeding sites based on reproductive success of their neighbours (*Boulinier et al., 2008*), migration of many animals can be described as 'informed dispersal' by which organisms choose when and to where they travel depending on environmental cues and stressors (*Clobert et al., 2009*). In contrast, movement of plants, which is mostly mediated by seed or fruit dispersal, is understood as a mostly passive process by which seeds or fruits are carried by biotic or abiotic vectors (*Bullock et al., 2006*; *Matthysen, 2012*). This contradicts the wide range of environmental sensing and responses observed in plant development and physiology. Is plant dispersal really that passive?

Wind is a particularly common plant dispersal vector. Though locally unpredictable, wind is globally prevalent and almost ubiquitous. Variation in diaspore morphology and structure can have significant impacts on wind dispersal distributions when comparing between species (*Muller-Landau et al., 2008*; *Tamme et al., 2014*) however, the relevance of morphological and structural variation *within* a species has been questioned for wind dispersal (*Greene and Johnson, 1992*; *Horn et al., 2001*; *Nathan et al., 2002*). In most cases, intraspecific variation in dispersal distances is primarily determined by environmental conditions rather than plant factors (*Nathan et al., 2002*; *Nathan et al., 2001*). For example, for tree samaras the coefficient of variation for samara terminal velocity was too small to have a significant effect on dispersal distance as variation in wind turbulence was far larger (*Greene and Johnson, 1992*). Additionally, in field conditions, there was no correlation between diaspore terminal velocity and the likelihood of uplifting (which is a prerequisite for long-distance dispersal) for *Liriodendron tulipifera* diaspores (*Horn et al., 2001*; *Nathan et al., 2002*).

However, some individual plant traits, such as plant height and diaspore appendages, can affect dispersal according to developmental programmes and in response to environmental cues (*Seale and Nakayama, 2020*). Bet-hedging strategies during dispersal can arise by manipulation of diaspore morphology (e.g. *Imbert, 2002*) and can be modified in response to nutrient availability, temperature, plant competition, or water availability (*Arshad et al., 2019*; *Imbert and Ronce, 2001*; *Martorell and Martínez-López, 2014*).

Modifying the timing of dispersal can also influence distances travelled. Phenological traits can ensure that plants produce and release seeds during seasonal periods that are favourable for dispersal (*Heydel and Tackenberg, 2017*; *Wright et al., 2008*). Abscission bias also affects localised timing of wind dispersal. By requiring a non-zero minimum force to detach diaspores from their parent plant, dispersal is often initiated only by greater wind speeds that result in longer flight times and further travelling distances (*Bohrer et al., 2008*; *Greene, 2005*; *Maurer et al., 2013*; *Pazos et al., 2013*; *Schippers and Jongejans, 2005*; *Soons and Bullock, 2008*; *Thompson and Katul, 2013*; *Treep et al., 2018*). For dandelions *Taraxacum officinale*, detachment arising from high horizontal wind speeds is thought to be important for the initiation of dispersal (*Greene, 2005*), whereas convective updrafts are likely to be important for long distance flight (*Tackenberg et al., 2003*).

Many wind dispersal processes rely on mechanical innovations to initiate dispersal or aid flight (*Seale and Nakayama, 2020*): for example, explosive dispersal via release of builtup pressure in the seed pod of *Cardamine hirsuta* (*Hofhuis et al., 2016*) or directed ejection of seeds to enable spinning during flight in *Ruellia ciliatiflora* (*Cooper et al., 2018*). Specific wing designs and weighting generate leading-edge vortices in autorotating samaras (*Lentink et al., 2009*), and pappus porosity allows the formation of a separated vortex ring (SVR) in dandelion diaspores (*Cummins et al., 2018*).

The dandelion SVR is a toroidal vortex that forms in the wake of the pappus generating a region of low pressure. This underlies the drag-dominated flight behaviour conferred by the pappus that allows the diaspore to fall both slowly and steadily through the air (*Cummins et al., 2018*; *Ledda et al., 2019*). SVR behaviour changes as the Reynolds number (*Re*) changes. *Re* is the ratio of inertial to viscous forces and dictates the behaviour of flow around an object. It depends on the characteristic length scale of the object (*D*), the velocity of fluid flow (*u*), and the kinematic viscosity of the fluid (*v*), and is calculated as

$Re = uD/v$. At Re >429, the SVR becomes unsteady and vortices are chaotically shed from the pappus (*Cummins et al., 2018*). At the natural falling velocity of dry dandelion diaspores, *Re* is approximately 355, which means that the SVR remains steady during flight. SVR stability is also affected by pappus porosity and thickness of the pappus (*Cummins et al., 2018*; *Ledda et al., 2019*).

So far, the SVR has been described for the dandelion pappus in their familiar open conformation. However, in response to high humidity, the dandelion pappus can morph into a closed structure by specific radial swelling of floral tissues (*Meng et al., 2016*; *Rothert, 1895*; *Seale et al., 2022*; *Taliev, 1894*). A ring of tissue, in which vascular tissue is surrounded by the cortical tissue, sits at the centre of the pappus and acts as a structure for the pappus hairs to attach to. Differential swelling capacities of the vasculature versus the cortical and other tissues facilitates raising and lowering of the pappus hairs (*Seale et al., 2022*). The presence of this specific mechanism and the widespread prevalence of pappus morphing in Asteraceae lineages (*Rothert, 1895*; *Taliev, 1894*) would suggest that pappus morphing serves a function.

To understand the biological significance of pappus morphing, we examined the fluid mechanics of the dandelion diaspore during flight and detachment from the parent plant. We found that, as the pappus changes shape, the SVR shrinks in size and moves closer to the pappus. Pappus closing also decreases the drag coefficient and thus increases diaspore falling velocity. When initiating dispersal, a closed pappus reduces detachment from the parent plant. We then used a computational model of wind dispersal to understand how these features affect dispersal distributions of the dandelion. Weather conditions combine with the functional consequences of pappus morphing to alter dispersal timing and flight behaviour and enhance dispersal distances compared to hypothetical diaspores that cannot morph.

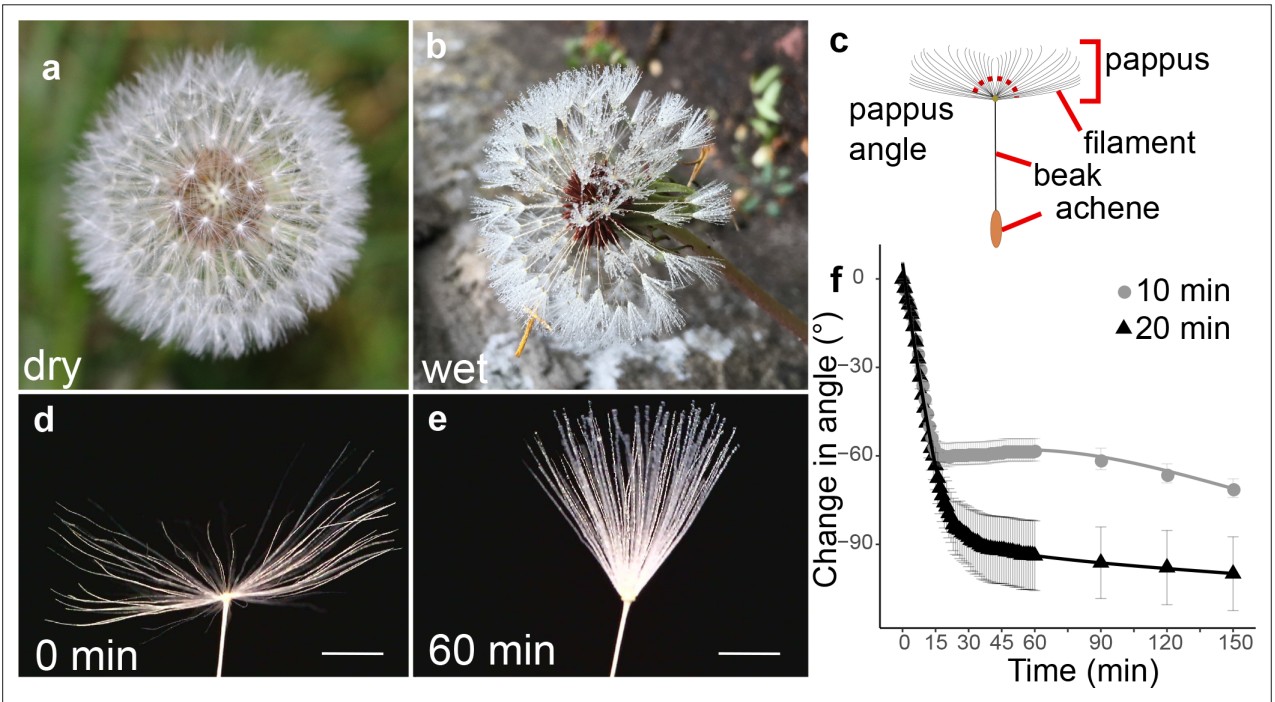

**Figure 1.** Moisture induces reversible closure of the dandelion pappus. (**a**) Image of a dry dandelion infructescence, (**b**) image of a wet dandelion infructescence, (**c**) schematic of a dandelion diaspore indicating features, and (**d**) image of dry dandelion pappus (scale bar: 2 mm). (**e**) Image of dandelion pappus after 1 hr in moisture chamber with a 20 min humidifier treatment (scale bar: 2 mm). (**f**) Time course of change in pappus angle between outermost hairs with 10 or 20 min humidifier treatment, n = 10 and n = 12 samples for 10 and 20 min treatments, respectively; error bars are s.e.m.

The online version of this article includes the following figure supplement(s) for figure 1:

**Figure supplement 1.** Angles of pappus hairs in dry and humid conditions.

# Results

The dandelion diaspore consists of an achene attached to a pappus comprising around 100 hair-like filaments (*Cummins et al., 2018*). The pappus is found closed in wetting conditions such as mist or rain (*Rothert, 1895*; *Taliev, 1894*; *Figure 1a–c*). In contrast, when dry, hairs are approximately horizontal, when measured from µCT scans at a 2 mm radius from the centre of the pappus (*Figure 1—figure supplement 1a*).

To capture pappus closure dynamics, we imaged dandelion diaspores in an enclosed, water-tight chamber with moisture added via an ultrasonic humidifier for 10 or 20 min (*Figure 1d–f*). For both treatments, the pappus angle (the angle between outermost hairs, see *Figure 1c*) rapidly decreased (4.0 ± 1.4 degree min⁻¹) during moisture treatments, which is consistent with previous experiments (*Meng et al., 2016*). This continued for 5–15 min after moisture addition ceased with a final change in angle of 60°–90° (*Figure 1f*). To test the effect of relative humidity (RH) below 100%, samples were placed in a chamber with stable RH. At RH of 71.8 and 87.0%, modest angles changes of around 10° and 24° were observed, respectively (*Figure 1—figure supplement 1b*). Together, these results indicate that the dandelion pappus partially closes at high RH and rapidly closes when fully wet.

We hypothesised that pappus shape changes would modify diaspore flight and indeed found a strong effect on the falling velocity of the diaspore (*Figure 2a*). A decrease in pappus angle of around 100–150°, associated with wetting for 1 hr, resulted in double to triple the falling velocity (*Figure 2a*). This is consistent with a change in projected pappus area that occurs when the pappus closes (*Figure 2—figure supplement 1a*). Additionally, a small increase in mass due to the added water resulted in a 0.7 µN increase in drag for the wetted samples.

To understand further how the pappus shape change affects the drag and falling velocity of the dandelion diaspore, we calculated the drag coefficient ($C_D$) and Reynolds numbers ($Re$) from our drop test data. The coefficient $C_D$ is a nondimensionalised parameter indicating the drag on an object for a given projected area (in our case, the area given by the pappus hairs). For dry diaspores, we examined $C_D$ and $Re$ alongside data from weighted/clipped diaspores from *Cummins et al., 2018* (*Figure 2b*). For dry diaspores, higher $Re$ is associated with rapidly decreasing $C_D$ before reaching a plateau. The wet diaspores with closed pappi, however, exhibit a different $C_D$-$Re$ relationship with substantially lower $C_D$ for a given $Re$ compared to dry (weighted) diaspores with open pappi (*Figure 2c*). Closing the pappus therefore has two simultaneous effects: $Re$ is increased as falling velocity increases, and the drag per unit of projected pappus area is decreased relative to a hypothetical open pappus of the same projected area. Therefore, pappus morphing is associated with shifting to a different region of the $C_D$-$Re$ parameter space (*Figure 2c*).

An SVR forms behind the pappus due to the high porosity of the pappus combined with precise spacing of the hairs, which limits air flow through the pappus. (*Cummins et al., 2018*; *Ledda et al., 2019*). The nature of the SVR behind circular disks is strongly affected by the porosity of the disk and its projected area. Closing the pappus has several effects on the geometry: firstly, the projected area exposed to the flow reduces (*Figure 2—figure supplement 1a*); secondly, porosity of the pappus reduces as hairs move closer together (*Figure 2c*); and thirdly, the shape of the pappus becomes less disk-like and more cone-like. We observed that the combination of these effects alters the fluid mechanics around the pappus by reducing vortex size and shifting the vortex towards the pappus (*Figure 2h and i*).

In order to understand how pappus morphing impacted on SVR formation, we examined the dandelion pappus in dry and wet states fixed in a vertical wind tunnel (*Figure 2d*). Flow visualisation confirmed the presence of the SVR in both open and partially closed pappi (*Figure 2e and f*). We performed particle image velocimetry (PIV) (*Raffel et al., 2018*) to spatially resolve the velocity field of air in the region behind the pappus. Streamwise flow velocity was kept constant to give Re = 133–197 across different pappus angles. Our PIV analysis was able to detect the expected features of the SVR, such as stagnation points in the flow and characteristic vorticity profiles. (*Figure 2g*, *Figure 2—figure supplement 1b*).

In the region of space directly behind the centre of the pappus, the maximum magnitude of negative streamwise flow velocity (minimum $u_z$) showed no clear relationship with the pappus angle (*Figure 2—figure supplement 1c*). However, the location of minimum $u_z$ correlated with pappus angle (*Figure 2g*). At more open pappus angles, the position of minimum $u_z$ moved further downstream of the pappus (*Figure 2g*, p=0.001, $R^2 = 0.34$). Additionally, the size of the vortex significantly

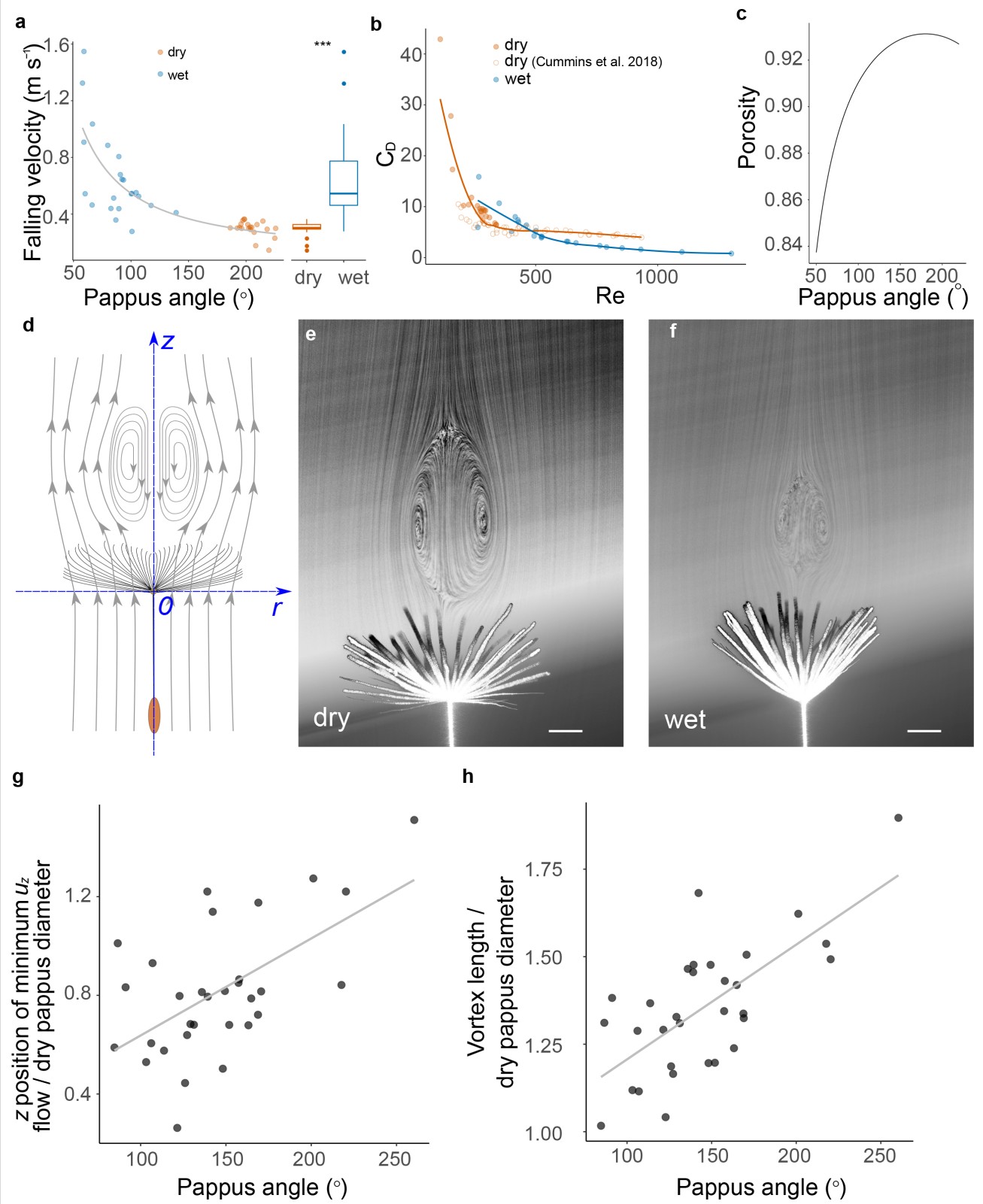

**Figure 2.** Pappus morphing alters diaspore flight and fluid dynamics. (**a**) Falling speed at varying pappus angles, before and after wetting for 1 hr in moisture chamber, asterisks indicate statistically significant difference at p<0.0001, n = 20; (**b**) relationship between Reynolds number (*Re*) and drag coefficient ($C_D$) for dry and wet samples, open red circles indicate data from weighted/clipped dry diaspores from ***Cummins et al., 2018***, n = 10 diaspores, filled circles are same data as (**a**), n = 20 diaspores; (**c**) estimated porosity at varying pappus angles; (**d**) schematic illustrating axes and

*Figure 2 continued on next page*

*Figure 2 continued*

direction of flow; (**e**) flow visualisation of dandelion fruit with open pappus, contrast adjusted to illustrate vortex more clearly (scale bar: 2 mm); (**f**) flow visualisation of dandelion fruit with a partially closed pappus, contrast adjusted to illustrate vortex more clearly (scale bar: 2 mm); (**g**) distance of the region of minimum $u_z$ from the pappus centre at varying pappus angle, position is nondimensionalised for each sample by the diameter of the pappus when dry, n = 12 diaspores; and (**h**) longitudinal length of the separated vortex ring (SVR) non-dimensionalised by the diameter of the pappus when dry, at varying pappus angles, n = 12 diaspores.

The online version of this article includes the following figure supplement(s) for figure 2:

**Figure supplement 1.** Changes to the diaspore falling velocity and separated vortex ring (SVR) as the pappus angle changes.

changes with altered pappus angle. The nominal length (see 'Materials and methods') of the vortex was markedly reduced at smaller pappus angles (*Figure 2i*, p<0.001, $R^2$ = 0.46).

From our data, it is clear that pappus morphology considerably affects the falling velocity of the diaspore and the fluid mechanics of the pappus. We also expected that the pappus shape might affect detachment of the diaspore from the parent plant. This is important because, as noted before, force thresholds on detachment are an important mechanism by which plants can bias timing of dispersal in favour of increased wind speeds (*Greene, 2005*; *Greene et al., 2008*; *Schippers and Jongejans, 2005*; *Skarpaas et al., 2006*; *Soons and Bullock, 2008*; *Thompson and Katul, 2013*; *Treep et al., 2018*) and is thought to promote long distance dispersal (*Schippers and Jongejans, 2005*; *Soons and Bullock, 2008*; *Thompson and Katul, 2013*). We placed dandelion infructescences in a wind tunnel (*Figure 3—figure supplement 1a*) and assessed the number of diaspores detaching at varying wind speeds (*Figure 3*). Hydrating samples for 1 hr by mist application significantly (p<0.001, $X^2$ = 341, df = 1) reduced detachment for the range of wind speeds tested (*Figure 3c*, *Figure 3—figure supplement 1b*). In flow with 9.1% turbulent intensity, 24% of dry diaspores remained attached at the maximum wind speed (9.8 m s⁻¹) compared to 50% of wet diaspores (*Figure 3c*). Reducing the turbulent intensity of the flow also reduced diaspore detachment (*Figure 3—figure supplement 1b, c*). These data indicate that hydration and air flow impact on the initiation of dispersal. This observation is consistent with detachment data for thistles (*Jongejans et al., 2007*; *Marchetto et al., 2012*; *Skarpaas et al., 2006*).

As drag, falling velocity, and diaspore detachment are strongly affected by moisture-dependent pappus morphing, we expected pappus closure would influence dispersal distances. We tested this hypothesis by numerically modelling the dispersal distance in dry or wet conditions corresponding to the open or closed state of the pappus. We first separately tested the effects of pappus closure on

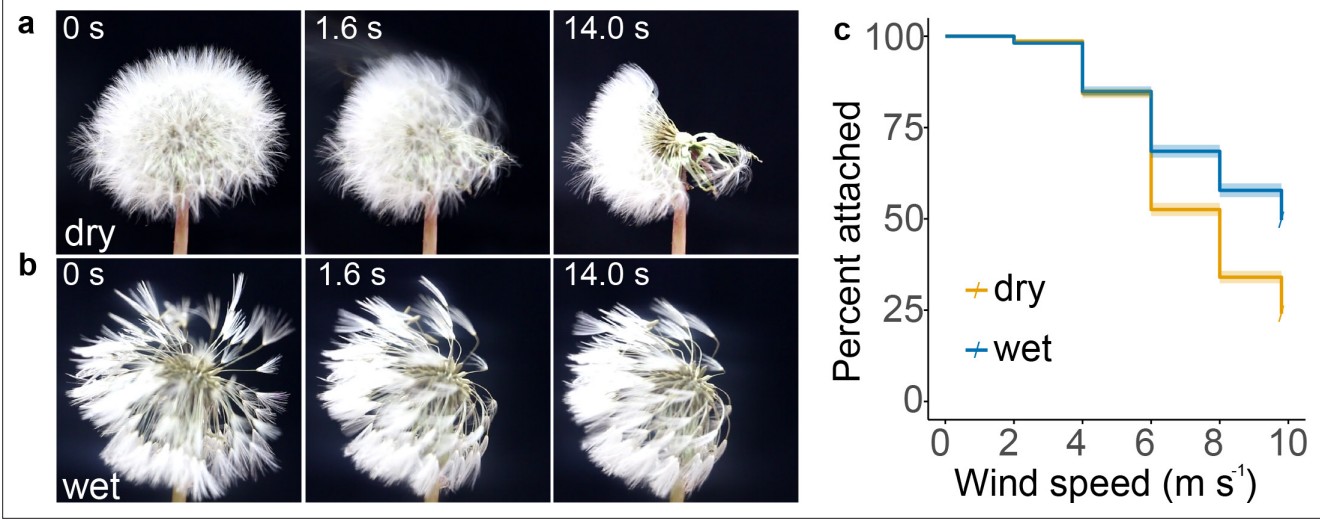

**Figure 3.** Detachment of diaspores from capitula. (**a**) Snapshots of detachment dynamics at 8 m s⁻¹ wind speed for a dry capitulum, air flow direction is from right to left of the images; (**b**) snapshots of detachment at 8 m s⁻¹ wind speed for a wet capitulum; and (**c**) survival plot of percentage of fruits attached to capitula at varying wind speeds for 9.1% turbulent intensity flow in dry and wet conditions, tested 10–14 days after opening of capitula, n = 20 capitula per treatment corresponding to 3001 and 2654 diaspores for dry and wet conditions, respectively; shading around lines indicates s.e.m.

The online version of this article includes the following figure supplement(s) for figure 3:

**Figure supplement 1.** Detachment assay setup and detachment measurements for young capitula and at varying turbulence intensity.

diaspore flight, then diaspore detachment, and finally combined these phenomena together to understand how pappus morphing affects dispersal.

We simulated wind dispersal trajectories using the Markov chain synthetic turbulence generation (STG) model previously developed and used for grassland species (*Pazos et al., 2013*; *Soons et al., 2004*). This model simulates diaspore dispersal according to instantaneous calculation of wind trajectories. Importantly, the model can be generalised to new settings using measured (or estimated) parameters and is able to capture long-distance dispersal in a realistic manner. Long-distance dispersal relies on uplifting of diaspores at early stages in the dispersal process and is primarily achieved within the model by incorporating autocorrelated horizontal and vertical turbulence generated from properties of the canopy vegetation. We used wind velocity values from the Edinburgh meteorological data separating it into two groups: dry conditions (RH < 90%) or wet conditions (RH ≥ 90%). There is a consistent negative correlation between wind speed and RH (*Figure 4a*, *Figure 4—figure supplement 1*). At high RH, when dandelion pappi are more likely to be closed, wind speeds are relatively low, although still variable (*Figure 4a*).

We ran five different model simulations to explore the different functional aspects of pappus morphology on dispersal. For each model, we simulated 10,000 dispersal events and computed kernel density estimates normalised such that the integral over distance equalled one. Firstly, we simulated dispersal of dandelion diaspores with open pappi in dry weather conditions (Model 1), when wind speeds tend to be higher. As expected, most diaspores would disperse a short distance (median: 4.3 m) from the parent plant, but the model generated dispersal distances with a right-skewed tail, indicating some diaspores will disperse much longer distances (*Figure 4c*). It is worth noting that extremely long distances (>10,000 km) are observed in a very small number of cases. Some, though not all, of these individual simulations occurred because the diaspores hit the top of the atmospheric boundary layer during trajectory simulation, but we found that omitting such individuals from the analysis made very little difference to the kernel density estimate. The accuracy of the model at this range cannot be verified due to the impracticality of tracking such events empirically. Without a clear rationale for a specific cut-off distance, we decided to leave the model distribution unchanged to avoid misrepresentation of the model's results.

To explore the effects of weather conditions alone on dispersal, we ran the model again using the wet weather wind speed distribution in a hypothetical scenario where the pappus remains open in these conditions (Model 2). This substantially decreased the median dispersal distance to 1.5 m (*Figure 4d*). It also decreased long-distance dispersers from 17.5% of simulated dispersal events in Model 1 to 5.6% in Model 2 (*Figure 4e*). This indicates that the lower wind speeds typical in wetter weather conditions (*Figure 4a*) would considerably decrease dispersal distances across the entire distribution even with the aerodynamically favourable open pappus (*Figure 4c–e*).

We then ran several simulations to investigate the ecological significance of closing the pappus. In Model 3, we used wet weather wind conditions, the falling velocity of the diaspore when the pappus is closed, and the probability of a given wind speed detaching the diaspore in the first place was based on the detachment profiles observed for wet infructescences (*Figure 3*). Compared to keeping the pappus open during wet weather, closing the pappus reduced median dispersal from 1.5 m to 0.9 m (*Figure 4c and d*). However, long-distance dispersal increased from 5.6% to 7.5% of the 10,000 simulated dispersal events (*Figure 4e*) giving an increase of 34%.

To understand how different aspects of pappus closing affect dispersal, we separately manipulated the detachment threshold probability and the falling velocity of the diaspore. Closing the pappus only during flight (Model 4) dramatically reduced median dispersal (4.3 m to 1.4 m), while keeping the pappus closed only during detachment (Model 5) considerably increased both median and long-distance dispersal (4.3 m to 6.6 m and 17.5% vs. 21.4% respectively).

Our model simulations established probability distributions for different realistic and hypothetical scenarios for diaspores that disperse. In reality, though, humidity may modify detachment thresholds to impact the probability of initiating dispersal in the first place. To explore this possibility, we simulated a 48 hr period during which dandelion plants are likely to disperse all their diaspores (*Treep et al., 2018*). Each instantiation of dispersal was considered to last 1 s, and diaspores were assessed for detachment against the appropriate probability distribution (derived from *Figure 3*) at the randomly selected wind speed for that second. There was a considerably higher probability of detaching in Models 1 and 3 when wind speeds were selected from the dry distributions and diaspores detachment

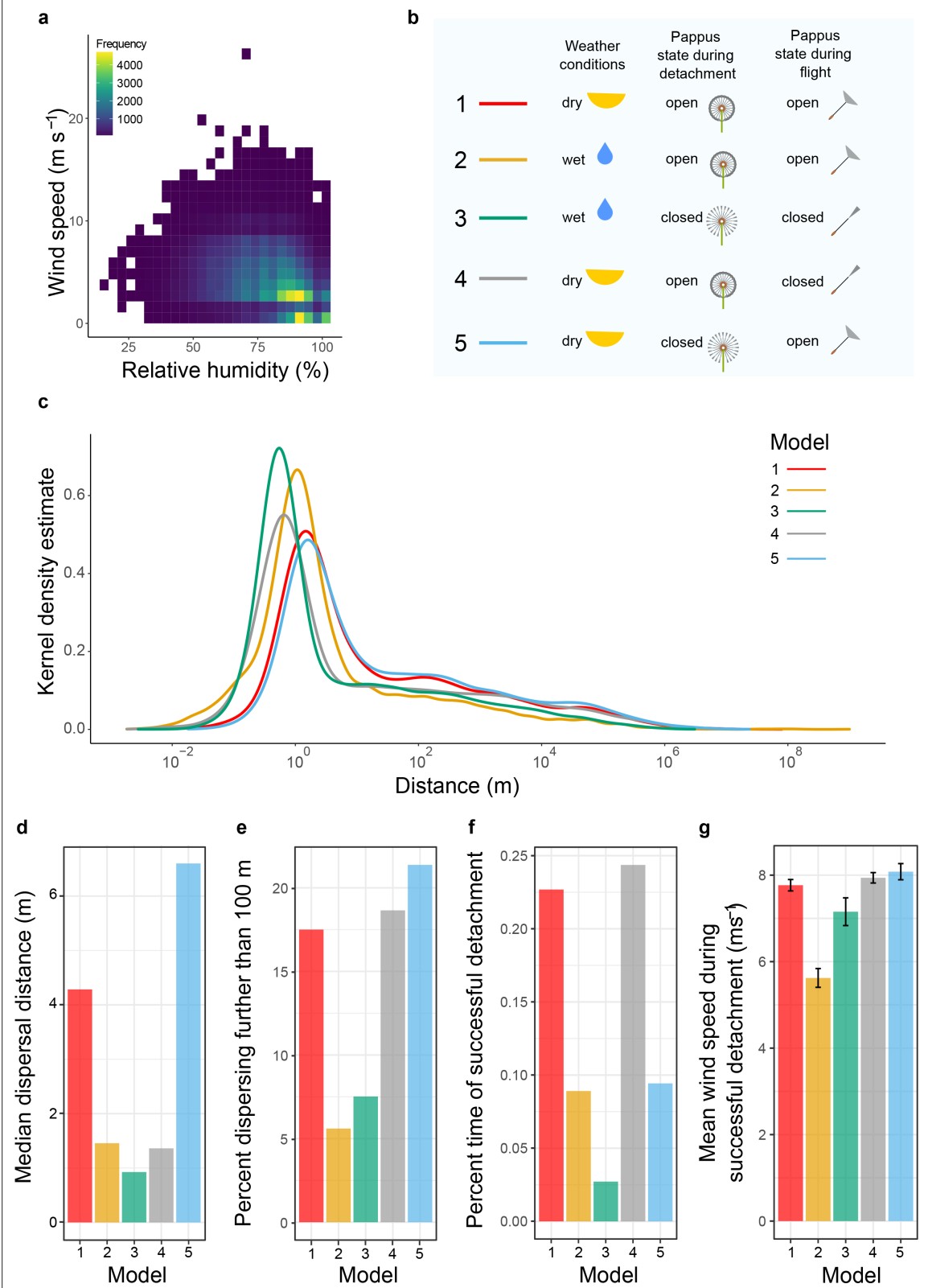

**Figure 4.** Change in pappus angle modifies dispersal according to meteorological conditions. (**a**) Relationship between relative humidity and wind speed in Edinburgh, UK. (**b**) Model descriptions indicating whether dry (relative humidity < 90%), or wet (relative humidity ≥ 90%) weather conditions were used, whether the pappus was simulated to be open or closed during detachment, and whether the simulated diaspore falling velocity during flight was based on an open or closed pappus. (**c**) Modelled kernel density estimate of predicted dispersal distances for diaspores with open or closed

*Figure 4 continued on next page*

*Figure 4 continued*

pappi and for different weather conditions (see **b** for model details). (**d**) Median dispersal distance for each model. (**e**) Percent of 10,0000 dispersal simulation events that travelled further than 100 m. (**f**) Percent of time in which diaspores successfully detached after 172,800 simulations (representing the number of seconds in a 48 hr period). (**g**) Mean wind speed for time points (172,800 simulations) when detachment occurred.

The online version of this article includes the following figure supplement(s) for figure 4:

**Figure supplement 1.** The relationship between relative humidity and wind speed in four additional locations.

**Figure supplement 2.** Modelled kernel density estimate of predicted dispersal distances for diaspores with a morphing or constantly open pappi in all weather conditions.

was based on having the pappus open (***Figure 4f***). When pappi were closed and weather was wet (Model 3), the probability of detachment dropped dramatically, which is likely due to the low wind speeds and higher detachment thresholds in these conditions. When detachment did occur, wind speeds were higher than when the pappi were open in wet weather (***Figure 4g***, Model 3 vs. Model 2). The data imply that dispersal in wet conditions is a rare event (***Figure 4f***), but that when it does occur, closing the pappus increases the chances of travelling long distances (***Figure 4e***, Model 3 vs. Model 2). We can conclude that closing the dandelion pappus maintains long-distance dispersal primarily via its effect on detachment (Model 4). This biases diaspore release towards dispersing in dry conditions when wind speeds tend to be higher (***Figure 4g***). Interestingly, for the few diaspores that do disperse in wet conditions, a greater spread of dispersal distances occurs with reduced median distances but slightly higher long-distance dispersal.

To understand the dispersal distribution of dandelion diaspores across all possible weather conditions, we combined the distribution of Models 1 and 3 weighted by the frequency of wet or dry weather conditions in our meteorological data, and also conditioned on the probability of detachment in such conditions (***Figure 4—figure supplement 2***). This kernel density estimate therefore represents the realistic dispersal distribution of diaspores given that they detach from the parent plant and that the pappus morphs depending on the humidity. We compared this to a hypothetical version in which the pappus cannot morph by combining Models 1 and 2 in the same manner. In our dataset, dry weather is three times more frequent than wet. Diaspores are 8 times less likely to detach in wet weather versus dry when morphing occurs and 2.5 times less likely when the pappus remains open. As a result dispersal during wet weather is rare in both morphing and open conditions so the distribution from Model 1 therefore dominated the overall kernel density estimates in both cases. To compare the models we found that the Jensen–Shannon divergence was 0.034 indicating that the two distributions follow essentially similar shapes (***Figure 4—figure supplement 2a and b***). Examining how the models varied, we found that morphing led to a 6% increase in long-distance dispersal (from 26.7% to 28.5% of seeds) (***Figure 4—figure supplement 2d***) and a 23% increase in median dispersal (from 3.9 m to 4.8 m) compared to a pappus that was always open (***Figure 4—figure supplement 2c***).

## Discussion

Structure is intimately linked to function in engineering and in biology. While biological systems have had millions of years of evolution to improve functional outcomes, evolution must also balance many survival and reproductive requirements which may lead to trade-offs. The dandelion diaspore achieves functionality during flight by constructing a filamentous pappus whose porosity allows an SVR to form. However, diaspore flight is not favourable in all circumstances, so we conjecture that the pappus morphs and modifies its structure to alter the flight characteristics.

As the pappus closes, the SVR decreases in size and shifts its position. Morphing via angular rotation of the pappus hairs is not only a convenient mechanism but also has functional consequences. If pappus hairs remained open but shortened in length, the drag per unit area would be greater than for a cone-like morphology with the same projected area. By generating a cone-like shape instead of a disk-like shape, the pappus becomes more streamlined. Together, the fluid mechanics of the closed pappus leads to significant changes in diaspore falling velocity and detachment from the parent plant.

Dispersing far from a parent is often considered favourable as resource and kin competition are reduced (***Saastamoinen et al., 2018***), but competition is not the only factor affecting site suitability. Soil moisture levels are temporally and spatially variable (***Vinnikov et al., 1996***), more than characteristics such as light and temperature (***Silvertown et al., 2015***), meaning that optimal germination sites

and times may vary considerably with changing weather and microenvironments. Our data suggest that dandelions have evolved pappus morphing because it biases detachment toward favourable dispersal times, which enhances both median and long-distance dispersal.

Moisture-induced pappus morphing may thus be a form of informed dispersal, by which dispersal is modified according to environmental cues signalling more favourable conditions (*Clobert et al., 2009*). In contrast to the popular perception that plant migration is a passive response to wind and other vectors, an explorative literature review has revealed otherwise (*Seale and Nakayama, 2020*). Plants are not able to make conscious decisions influencing their behaviour, but can form mechanical or chemical structures that respond differentially to the environment. There are two types of informed dispersal in plants: developmental adaptation of heteromorphic diaspores to environmental stresses (*Martorell and Martínez-López, 2014*), which acts over a longer time scale responding to stable environmental conditions; and highly temporal and local responses such as the fire-induced dispersal of *Banksia* (*He et al., 2011*), and the temperature-controlled dehiscence of *Brassica* siliques (*Li et al., 2018*), as well as the dynamic pappus morphing observed here. Such short-term regulation may be particularly beneficial in responding to unpredictable or fast-changing conditions.

The effects of pappus closure on detachment and on flight counteract one another to some extent as diaspores that detach when wet are more likely to do so when wind speeds are higher, but a closed pappus will cause them to fall faster. However, the low likelihood of detaching at low windspeeds when dry (due to higher average wind speeds) and when wet (due to pappus closure) means that dispersal largely occurs when wind speeds are high. These higher wind speeds appear to be more significant than the change in falling velocity as the overall effect of a morphing pappus is to increase median and long-distance dispersal (*Figure 4*, S5). The effect on long-distance dispersal is particularly significant as this has a disproportionate effect on ecological distributions and is often the major driver of range expansion (*Clark, 1998*; *Kot et al., 1996*; *Trakhtenbrot et al., 2005*).

Pappus morphing may have additional functions beyond detachment, flight, and dispersal, such as protection from damage, or to aid germination. The morphing phenomenon occurs in many (perhaps most) pappus-bearing diaspores but is conspicuously missing from some representatives of the Aster-aceae (e.g. *Leontodon, Hieracium, Erigeron*) (*Rothert, 1895*; *Taliev, 1894*). A comparative approach examining the ecological and phenological features of pappose fruits that do or do not morph may yield greater insight into the roles of pappus morphing.

We have tested the ecological implications of this biomechanical response to moisture and suggested that pappus closure acts to tune dispersal promoting long-distance dispersal by responding to moisture conditions. As many species modify their seed or fruit structure in response to water, this might be a general mechanism to ensure dispersal at appropriate times.

Considering how prevalent environmentally induced plasticity is in plant structure and its development, understanding of structural functions within environmental contexts is critical to predict the ecological consequences of climate change (*Denny and Helmuth, 2009*). Insights into ecological and environmental drivers in turn enable us to comprehend selective pressure that has acted and will act to evolve plant forms and their functions.

## Materials and methods
### Plant growth and samples

*Taraxacum officinale* agg. diaspores were collected and grown as described previously (*Cummins et al., 2018*). Briefly, plants were collected in Edinburgh, UK (55.922684°N, 3.170703°W), and grown in a peat/sand mix compost in greenhouses supplemented with light where necessary to ensure a 16 hr day, 8 hr night and kept at 18–21°C. All samples were the progeny of plants grown in the greenhouse for two generations originating from the same individual. As this subspecies reproduces apomictically, all diaspores used were considered to be genetically identical.

For the detachment assay, stems with capitula were harvested once the inflorescence had closed but before the infructescence (capitulum) had opened. Polypropylene tubes of 15 ml capacity were filled with water and covered by a piece of parafilm with a small hole in. Stems of samples placed through the hole. These samples were transported to the site of the wind tunnel experiment and then placed in a greenhouse (ambient conditions except for day length, which is artificially extended with

electric lighting to ensure at least 16 hr of daylight) to allow the infructescence to fully mature and open.

## Pappus angle measurements

Angles of each hair from the horizontal axis were calculated at a 2 mm radius from the centre of the pappus, with the beak aligned to the vertical axis. Source data and analysis of 3D micro-tomography CT scans were identical to the ones used in *Cummins et al., 2018*.

For all other pappus angle measurements, the angle was measured from 2D microscope or macro-camera images. The pappus was aligned perpendicular to the objective/lens and pappus angle measured as the angle between the outermost hairs.

## Moisture chamber imaging

The bespoke moisture chamber consisted of a 70 l airtight plastic box (Solent Plastics, UK). A hole was made in the box to pass cables through and the space around sealed with silicone sealant. Two USB microscopes (Maozua, USB001) were positioned horizontally to image dandelion diaspore samples, which were fixed in place by embedding in plasticine or individual pieces of foil shaped around the achene. An ultrasonic humidifier (Bottle Caps) was filled with distilled water and placed next to the dandelion samples. The ultrasonic humidifier releases small droplets of liquid water into the chamber meaning that RH reaches 100% within the first few minutes. Samples were imaged each minute for the duration of the experiment.

For RHs < 100%, dishes of saturated salt solutions were placed into the chamber to maintain defined, stable humidity levels (*Wexler and Hasegawa, 1954*). RH was monitored with a datalogger (Lascar, Easylog EL-GFX-2). Samples were imaged at 0 hr and 10 hr when RH had long reached equilibrium.

## Falling velocity assays

Drop tests were carried out using a modified version of the protocol described in *Cummins et al., 2018*. Diaspore samples were tested in two batches (n = 10 and n = 11). First, all diaspores were weighed together to obtain the mean weight per sample. Each sample was dropped three times and video recorded with a DSLR camera (Canon EOS 70D). Samples were then imaged in the moisture chamber while still dry. The humidifier was switched on for 1 hr and samples re-imaged. Diaspores were removed from the chamber one by one and dropped three times before placing back inside the humid chamber to prevent further drying. After all samples were dropped, all were weighed together again. The mean increase in weight after wetting was 70 μg, which corresponds to approximately 10% greater weight when wet compared to dry. Falling velocity was determined from the video frames using a particle detection script (*Cummins et al., 2018*).

## Calculated variables for fluid mechanics analysis

The projected area of the pappus was calculated according to the pappus angle, assuming 100 straight, non-overlapping hairs of length 7.41 mm and diameter 16 μm according to previous measurements (*Cummins et al., 2018*). The apical plate at the centre of the structure (where the hairs attach) was not included in these calculations. Porosity was calculated as the ratio of empty space between hairs (i.e. 1 – the projected area) to the total area of the circle that would enclose the projected area.

The Reynolds number is defined as $Re = uD/\nu$, where $u$ is the flow velocity (the falling velocity, that is, the undisturbed velocity of the fluid with respect to the pappus), $D$ is the characteristic length scale (in our case, this is the diameter of the pappus assuming hair length of 7.41 mm and varies according to pappus angle), and $\nu$ is the kinematic viscosity of the fluid at 20°C for which we used $15.06 \times 10^{-6}$ m$^2$ s$^{-1}$.

The drag coefficient was calculated as follows:

$$C_D = \frac{mg}{\frac{1}{2}\rho u^2 A}$$

where $\rho$ = 1.0241 kg m$^{-3}$ is the density of air, $A$ is the projected area of the pappus, taking the porosity into account, and $g$ = 9.81 m s$^{-1}$ is gravitational acceleration. Additionally, $m$ is the mass of the

diaspore, for which a mean measured value of 0.614 mg was used for all dry samples and 0.684 mg for all wet samples.

## PIV and flow visualisation

Samples were fixed in place inside a vertical wind tunnel for flow visualisation and PIV (*Cummins et al., 2018*). Long-exposure images were obtained using a Canon DSLR camera (EOS 70D). The raw experimental images were processed to distinguish the region of interest (i.e. the SVR) from the reflective pappus. To do this, we applied a digital graduated neutral density filter using the 'Curves' contrast adjust tool and the 'Blend' tool within the open-source software GNU Image Manipulation Program (GIMP version 2.8.22).

For PIV, videos were obtained with a high-speed camera (Fastcam Photron SA1.1) shooting at 125 frames per second fitted with a macro lens (Tamron). Air flow velocity was 0.207 m s$^{-1}$ throughout.

For each video, 100 frames were analysed in PIVlab (MATLAB; *Thielicke and Stamhuis, 2014*) using single-pass direct cross-correlation with a window size of 128 pixels (corresponding to 3.88 mm) and a window overlap of 64 pixels. Data were filtered by excluding vectors of more than 2 standard deviations and a normalised median filter (*Westerweel and Scarano, 2005*) with a minimum normalisation level ($\epsilon$) of 0.1 pixels and a detection threshold of 3. Vectors were interpolated where missing, and the mean vector field of all 50 pairs of images was calculated.

Noisy rows of data from the images were excluded from further analysis. The first row of the image (row size defined by the window size), which was always noisy, was removed. Other noisy rows were identified by fitting an autoregressive integrated moving-average model (R package 'tsoutliers') to the profiles of streamwise velocity ($u_z$) across the $r$ direction of the image. The velocity profiles with outliers were considered noisy. A $z$ portion of the image with no more than one consecutive noisy row was considered to form the vortex region and was used for further analysis. Analysis was also limited in the $r$ direction by excluding regions of data that fell outside of the limits of the dandelion pappus width. The coordinates of the central point of the pappus and the pappus angle were measured from images. The pappus limits were calculated assuming the pappus formed a triangular shape with hairs that were 7.41 mm long (*Cummins et al., 2018*).

The point $P$ of minimal flow was determined by finding where $u_z$ along the $z$-axis was minimal at the $r$ location of the centre of the pappus. Let $z_p$ be the $z$-coordinate of $P$, $D$ the diameter of the pappus and $\tau$ the pappus angle. We found that

$$\frac{z_p}{D} = 0.24 + 0.0039\,\tau,$$

p=0.001, $R^2 = 0.34$.

Using this relationship (*Figure 2f*), we identified a nominal $z$-coordinate of minimal $u_z$.

The flow structure was not well resolved beyond the limits of the vortex, so it was not possible to identify the downstream stagnation point where the velocity vanishes. Instead, the nominal vortex length was determined by obtaining the point beyond the centre of the pappus at which $u_z = 0.02$ m s$^{-1}$. This does not give the actual length of the vortex but gives an indicative value to compare between samples and treatments.

## Detachment assays

A modified version of the method in *Jongejans et al., 2007* was used to assess detachment of diaspores from the capitulum. Tests were conducted in a bespoke horizontal, open-circuit wind tunnel set up in a greenhouse at the University of Glasgow, UK (*Zhdanov et al., 2021*). The test section was 1 m long with a rectangular cross-section of 450 mm × 400 mm (*Figure 3—figure supplement 1a*). Either 3 days (*Figure 3—figure supplement 1b, c*) or 10–14 days after opening (*Figure 3c*), samples were placed in the middle of the test section and were subjected to a flow speed of 2 m s$^{-1}$ for 30 s. Samples were held in place by passing an aluminium rod (rod diameter was 1.5 mm or 3 mm in length depending on stem diameter) through the hollow stem attached to the capitulum. For older samples, in which the stem had degraded, stem remnants were clamped to the rod. Detached diaspores were collected in a mesh at the end of the test section (*Figure 3—figure supplement 1a*) and counted. This was repeated for 4, 6, 8 and 10 m s$^{-1}$.

After the maximum wind speed, the number of diaspores remaining on the capitulum was also counted. For more turbulent conditions, a grid consisting of 11 equally spaced vertical rods (with a

diameter of 10 mm) was created and placed inside the test section (200 mm downstream of the inlet). This increased turbulence intensity to 9.1%, but the maximum wind speed was reduced slightly to 9.8 m s$^{-1}$, and this was used for the maximum detachment measurement speed instead of 10 m s$^{-1}$. The turbulent intensities used here are comparable to that in some natural conditions (*Ghanadi et al., 2017*), though turbulence in the atmospheric boundary layer exhibits enormous variability, with values as high as 400% reported (*Wilson et al., 1982*).

Wet capitula were treated by placing in the moisture chamber with the humidifier switched on for 1 hr before placing in the wind tunnel.

## Meteorological data

Hourly meteorological data was obtained from the Integrated Surface Database from the National Centers for Environmental Information (USA). We compared meteorological data from five locations across the UK (Edinburgh, Manchester, Nottingham, Norwich, and Exeter) and found the wind speed profiles to be largely similar (Figure S4). For all further analysis and modelling, we used the meteorological data from Edinburgh. Hourly data for a 30-year period (1987–2017) were acquired for Edinburgh (55.950–3.373) and filtered according the following criteria: entries were excluded if hourly wind speed, dewpoint, or temperature were missing; data were only used from the beginning of April to the end of October in each year (the period of dandelion reproduction); occasional data measured between hours were excluded. RH was calculated from the dewpoint and temperature. The hourly wind speeds varied around a mean of 4.0 m s$^{-1}$ (*Figure 4a*; measured 10 m reference height). RH ranged between 14 and 100% with a mean value of 80%.

## Dispersal modelling

Using the above meteorological data, we simulated the dispersal of dandelion diaspores using a mechanistic trajectory model (the CELC Markov chain STG model; details as in *Soons et al., 2004*). This model simulates the dispersal of individual diaspores based on mean horizontal wind speeds, vegetation characteristics, and plant traits, and a resulting modelled vertical wind speed and turbulence distribution. Wind speeds at infructescence height were inferred as described by *Soons et al., 2004*, assuming a logarithmic vertical wind profile.

For a representative environmental setting, we assumed the release height of dandelion diaspores to be 0.35 m (*Treep et al., 2018*) and surrounding vegetation height to be 0.2 m with a leaf area index of 2.0 (*Soons and Bullock, 2008*). We simulated dispersal under dry (RH < 90%) and wet conditions (RH > 90%) using the respective wind speed distributions. As dandelion pappi can partially close at RH > 70% (*Figure 1—figure supplement 1b*) and field-measured RH > 90% is a good proxy for wetness on plant surfaces (*Sentelhas et al., 2008*), we used RH = 90% as a cut-off point. Diaspore falling velocities for dry and wet conditions were 0.3 m s$^{-1}$ and 0.7 m s$^{-1}$, respectively, corresponding to the measured values from our drop tests.

Detachment probabilities for dry and wet weather were based on the wind tunnel detachment assays described above. Alternative parametric models were fitted to the detachment data using the R package 'flexsurv' (*Jackson, 2016*). According to the Akaike information criteria (AIC), a generalised gamma model fit the data best and performed better than Weibull, exponential, log-normal, log-logistic, generalised F, and Gompertz models. The fitted generalised gamma model was extrapolated beyond the original limit of 10 m s$^{-1}$, and the probability of detachment for each humidity condition was used in the implementation of the dispersal model simulations.

## Data analysis and statistics

All measurements were carried out on separate individual diaspores from different infructescences except detachment assays, in which multiple diaspores from each infructescence were counted.

Detachment assays were analysed by fitting Kaplan–Meier survival models to the data, with log-rank tests used to compare treatments (R package 'survival'). For falling velocity tests, the following model was fitted to the data where the falling velocity is proportional to $\cos^{-2/3} \tau$, similarly to *Meng et al., 2016*, and dry and wet groups were compared using Student's two-sided two-sample *t*-test. For PIV data, linear regression models were fit to the data. For the plot of the relationship between $C_D$ and $Re$, loess regression with a span of 0.8 was used to illustrate the trends in the data. For boxplots,

the centre line is the median, hinges indicate first and third quartiles, and whiskers extend to largest value no further than 1.5 times the interquartile range.

## Acknowledgements

We thank Alex Twyford, Jonathan Silverton, and Richard Milne for helpful discussion on the project and manuscript. This work was funded by Leverhulme Trust (RPG-2016-255) to IMV, EM, and NN and (ECF-2019-424) to MS. MRB was funded by the BBSRC (P011586/1 and T006153/1); JMB under UKCEH National Capability project 06895; IVM by the European Research Council (H2020 ERC-2020-COG 101001499); and NN by the Royal Society (UF140640 and URF\R\201035).

## Additional information

### Funding

| Funder | Grant reference number | Author |
| --- | --- | --- |
| Leverhulme Trust | RPG-2016-255 | Enrico Mastropaolo Ignazio M Viola Naomi Nakayama |
| Leverhulme Trust | ECF-2019-424 | Madeleine Seale |
| Biotechnology and Biological Sciences Research Council | P011586/1 and T006153/1 | Michael R Blatt |
| European Commission | ERC-2020-COG 101001499 | Ignazio M Viola |
| Royal Society | UF140640 and URF-R-201035 | Naomi Nakayama |

The funders had no role in study design, data collection and interpretation, or the decision to submit the work for publication.

### Author contributions

Madeleine Seale, Data curation, Formal analysis, Supervision, Funding acquisition, Validation, Investigation, Visualization, Methodology, Writing – original draft, Writing – review and editing; Oleksandr Zhdanov, Formal analysis, Investigation, Methodology, Writing – review and editing; Merel B Soons, Software, Formal analysis, Investigation, Methodology, Writing – original draft, Writing – review and editing; Cathal Cummins, Formal analysis, Methodology, Writing – review and editing; Erika Kroll, Formal analysis, Investigation, Writing – review and editing; Michael R Blatt, Resources, Formal analysis, Supervision, Funding acquisition, Methodology, Project administration, Writing – review and editing; Hossein Zare-Behtash, Angela Busse, Resources, Formal analysis, Supervision, Methodology, Writing – review and editing; Enrico Mastropaolo, Conceptualization, Formal analysis, Supervision, Funding acquisition, Project administration; James M Bullock, Formal analysis, Funding acquisition, Methodology, Writing – review and editing; Ignazio M Viola, Conceptualization, Formal analysis, Supervision, Funding acquisition, Methodology, Project administration, Writing – review and editing; Naomi Nakayama, Conceptualization, Resources, Formal analysis, Supervision, Funding acquisition, Visualization, Methodology, Writing – original draft, Project administration, Writing – review and editing

### Author ORCIDs

Oleksandr Zhdanov (iD) http://orcid.org/0000-0002-1742-9765
Erika Kroll (iD) http://orcid.org/0000-0001-8832-7208
Ignazio M Viola (iD) http://orcid.org/0000-0002-3831-8423
Naomi Nakayama (iD) http://orcid.org/0000-0002-9390-3545

### Decision letter and Author response

Decision letter https://doi.org/10.7554/eLife.81962.sa1
Author response https://doi.org/10.7554/eLife.81962.sa2

## Additional files

### Supplementary files
• Transparent reporting form

### Data availability
The Source data for all figures has been deposited to Zenodo, doi: https://doi.org/10.5281/zenodo.7038366.

The following dataset was generated:

| Author(s) | Year | Dataset title | Dataset URL | Database and Identifier |
| --- | --- | --- | --- | --- |
| Seale M | 2022 | Environmental morphing enables informed dispersal of the dandelion diaspore | https://doi.org/10.5281/zenodo.7038365 | Zenodo, 10.5281/zenodo.7038366 |

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
