## [Editor Report]

This boundary-crossing work on dandelion diaspore flight is an excellent demonstration of how to address fundamental questions about wind dispersal of plant seeds from biophysical and ecological perspectives. Both wind-tunnel experiments and models provide compelling evidence that the aerodynamics of dandelion diaspores change with the environment. Addition of local climate data enables the authors to make a convincing case about how the biophysical properties can scale up to affect dispersal across the landscape under different environmental conditions. In addition to the strong data, this is a clear, accessible, and very enjoyable read.

---

## [Decision Letter]

**Decision letter after peer review:**

Thank you for submitting your article "Environmental morphing enables informed dispersal of the dandelion diaspore" for consideration by *eLife*. Your article has been reviewed by 2 peer reviewers, and the evaluation has been overseen by a Reviewing Editor and Meredith Schuman as the Senior Editor. The following individuals involved in review of your submission have agreed to reveal their identity: Frederick P. Gosselin (Reviewer #1); Lauren Sullivan (Reviewer #2).

The work was received very favorable and thus there are mostly cosmetic changes to be completed before this is appropriate for publication.

Essential revisions:

1) Address the confusion about wet weather predictions from Model 5 (see Rev 2 comments).

2) Reassess the model in Figure S5 in light of comments from reviewer 1.

3) Change colors in Figures 1-2 to improve readability (Rev 2).

*Reviewer #1 (Recommendations for the authors):*

Figure S5 of the modelled kernel of density of the predicted flight distance seems underwhelming. The two curves are highly similar and it's difficult to assess if the difference is significant on the logarithmic scale. Could a statistical test be performed to provide a quantification of how different they are? Similarly, since weather statistics is playing a dominant role, could the influence of pappus morphing be assessed with a principal component analysis or some other tool? Basically, how impactful is the dry/windy probability versus morphing/non-morphing effect in the final kernel?

*Reviewer #2 (Recommendations for the authors):*

The paragraph that starts on line 315 is really key to the ecological understanding of the work, yet is a bit confusing for me. For example, the authors say "When pappi were closed and weather was wet (Model 5)" that this increases dispersal ability. However, model 5 from the figure indicates that the weather is dry. I think this paragraph could be reworked to be more clear as to why the authors think that wet weather promotes longer dispersal, as from what I see, the models 2 and 3 appear to be the ones with wet weather, and also seem to have PDF's that don't indicate as far of travel (Figure 4b-f). The authors state that the increased wind is the cause, but I do not see the evidence for this in this paragraph or in Figure 4.

Another weakness to me, however an easy to fix weakness, is that in Figures 1g,h and 2c the red to blue gradient of points is unclear to me. I don't think it's necessary. I prefer to leave the points black, and keep the blue/red difference for the dry and wet seeds. In addition I'd encourage the authors to select two colors that are more visually different for those who are colorblind. Also, whatever colors the authors choose for dry/wet conditions, I'd keep those the same throughout the main figures, and the supplemental figures (the authors switch to yellow/blue in the supplement).

Something about the wording on lines 316-317 is confusing to me. It seems like it should read that humidity doesn't matter at all for dispersal. Anyway, if you could reword this sentence that would help!

---

## [Author Response]

Reviewer #1 (Recommendations for the authors):Figure S5 of the modelled kernel of density of the predicted flight distance seems underwhelming. The two curves are highly similar and it's difficult to assess if the difference is significant on the logarithmic scale. Could a statistical test be performed to provide a quantification of how different they are? Similarly, since weather statistics is playing a dominant role, could the influence of pappus morphing be assessed with a principal component analysis or some other tool? Basically, how impactful is the dry/windy probability versus morphing/non-morphing effect in the final kernel?

Thank you for your comments. As the kernels in Figure S5 (now labelled Figure 4—figure supplement 2) arise from our modelling rather than empirical data, we could not meaningfully carry out inferential statistical tests. Instead, to address this issue, we have computed the Jensen-Shannon divergence metric (0.034), which gives a quantitative indication of the difference between the models. The result indicated that the two models have similar shapes but that there are some quantitative differences between them. We have highlighted this in lines 345-349.

In terms of breaking down the model into its component parts, we have partially done this in Figure 4 with the different models (varying weather, pappus flight, pappus detachment) indicating the effects of each of these elements. Three of these models (1-3) were directly used to construct Figure 4—figure supplement 2 with different weightings according to the frequency of weather types and dispersal probability. The frequency of detachment beyond the threshold for wet conditions was very low; therefore, one of them (1) dominantly affected the graphs shown in Figure 4—figure supplement 2. To make it clearer what the contributions of each of these component models was, we have added quantitative values in the text (lines 334-339) indicating the probabilities of the weather and detachment to explain how the final distribution was reached. We have also added in some additional panels to Figure 4—figure supplement 2 to visually show the numbers more clearly. We hope that this will aid understanding of this part of the work.

Reviewer #2 (Recommendations for the authors):The paragraph that starts on line 315 is really key to the ecological understanding of the work, yet is a bit confusing for me. For example, the authors say "When pappi were closed and weather was wet (Model 5)" that this increases dispersal ability. However, model 5 from the figure indicates that the weather is dry.

Thank you for spotting this. This was, in fact, an error made when the manuscript was being edited. We meant to refer to Model 3 here, not Model 5 and this is now corrected.

I think this paragraph could be reworked to be more clear as to why the authors think that wet weather promotes longer dispersal, as from what I see, the models 2 and 3 appear to be the ones with wet weather, and also seem to have PDF's that don't indicate as far of travel (Figure 4b-f). The authors state that the increased wind is the cause, but I do not see the evidence for this in this paragraph or in Figure 4.

Yes, we agree that this could be made clearer. We have improved the wording to make it more specific what we are referring to and have added an additional figure 4g to show the wind speed during successful detachment. We hope this helps with understanding.

Another weakness to me, however an easy to fix weakness, is that in Figures 1g,h and 2c the red to blue gradient of points is unclear to me. I don't think it's necessary. I prefer to leave the points black, and keep the blue/red difference for the dry and wet seeds. In addition I'd encourage the authors to select two colors that are more visually different for those who are colorblind. Also, whatever colors the authors choose for dry/wet conditions, I'd keep those the same throughout the main figures, and the supplemental figures (the authors switch to yellow/blue in the supplement).

Thanks for pointing this out. We have modified the colours in the figures as suggested and stuck with the orange/blue scheme, which should be suitable for colourblind readers.

Something about the wording on lines 316-317 is confusing to me. It seems like it should read that humidity doesn't matter at all for dispersal. Anyway, if you could reword this sentence that would help!

We have modified this sentence to make it clearer.